# Improvement in Nitrogen-Use Efficiency Increases Salt Stress Tolerance in Rice Seedlings and Grain Yield in Salinized Soil

**DOI:** 10.3390/plants14040556

**Published:** 2025-02-11

**Authors:** Ping Ji, Chen Xu, Fenglou Ling, Xingjie Li, Zexin Qi, Yunfeng Chen, Xiaolong Liu, Zhian Zhang, Jinze Wang, Zhiyang Luo, Ziwen Cheng, Jianrui Chen

**Affiliations:** 1College of Life Sciences and Resources and Environment, Yichun University, Yichun 336000, China; 190116@jxycu.edu.cn (P.J.);; 2Institute of Agricultural Resources and Environment, Jilin Academy of Agriculture Sciences, Changchun 130033, China; 20160939@mails.jlau.edu.cn; 3Faculty of Agronomy, Jilin Agricultural University, Changchun 130118, China

**Keywords:** rice (*Oryza sativa* L.), salt stress, nitrogen-use efficiency, grain yield

## Abstract

Salt stress has become a major limiting factor of rice (*Oryza sativa* L.) yield worldwide. Appropriate nitrogen application contributes to improvement in the salt tolerance of rice. Here, we show that improvement in nitrogen-use efficiency increases salt stress tolerance in rice. Rice varieties with different nitrogen-use efficiencies were subjected to salt stress; they were stimulated with 50, 100, and 150 mmol/L of NaCl solution at the seedling stage and subjected to salinities of 0.2, 0.4%, and 0.6% at the reproductive growth stage. Compared with nitrogen-inefficient rice varieties, the nitrogen-efficient rice varieties showed significant increases in the expression levels of nitrogen-use-efficiency-related genes (*TOND1* and *OsNPF6.1*), nitrogen content (5.1–12.1%), and nitrogen-use enzyme activities (11.7–36.4%) when under salt stress conditions. The nitrogen-efficient rice varieties showed a better adaptation to salt stress, as shown by the decrease in leaf-withering rate (4.7–10.3%), the higher chlorophyll (3.8–9.7%) and water contents (1.1–9.2%), and the better root status (7.3–9.1%) found in the rice seedlings under salt stress conditions. Analysis of physiological indexes revealed that the nitrogen-efficient rice varieties accumulated higher osmotic adjustment substances (9.7–79.9%), lower ROS (23.1–190.8%) and Na^+^ (15.9–97.5%) contents, higher expression levels of salt stress-related genes in rice seedlings under salt stress conditions. Furthermore, the nitrogen-efficient rice varieties showed higher yield under salt stress, as shown by a lower salt-induced decrease in 1000-grain weight (2.1–6.2%), harvest index (1.4–4.9%), and grain yield (2.8–4.1%) at the reproductive growth stage in salinized soil. Conversely, the nitrogen-efficient rice varieties showed better growth and physiological metabolism statuses under severe salt stress conditions. Our results suggest that nitrogen-efficient rice varieties could improve nitrogen-use and transport efficiency; accordingly, their use can improve the gene expression network, alleviating salt damage and improving grain yield under severe salt stress conditions.

## 1. Introduction

Soil salinization is a severe environmental and agricultural threat globally. According to the Food and Agriculture Organization (FAO), over 950 million ha of land suffers from soil salinization globally [1]. With the development of industry and the impact of human activities, the area of salinized soil is expanding, causing severe yield loss for crops worldwide [2,3]. It is projected that the world’s population will reach 9.7 billion by the year 2050, with a 60% increase in crop yield requirements compared with the requirements of the present population [4]. Thus, the rational utilization of salinized soil for improved crop production remains a huge challenge worldwide.

Salt stress is one of the major limiting abiotic factors for crop production globally. Plants growing in salinized soil suffer from continuous osmotic stress [5], serious ion toxicity [6], and oxidative stress induced by excess reactive oxygen species (ROS) [7,8]. Salt stress has the following effects: it inhibits plant growth [9]; it decreases photosynthetic efficiency and chlorophyll fluorescence [10]; it destroys plants’ osmotic adjustment systems, antioxidant defense systems, and ionic balances [5,11]; it affects nutrient uptake [12]; finally, it suppress crop yields [13,14]. These impacts make it imperative to explore ways to improve crop yield under salinized soil conditions.

Rice is one of the most important food crops worldwide, feeding more than half the world’s population. However, rice is moderately sensitive to salt stress [15]. High salinity in soil inhibits the growth of rice plants [16], resulting in physiological water loss, ion toxicity, and membrane injury [17,18]; leading to biomass loss or pollen abortion [19]; and finally, causing yield loss [20]. Rice shows different tolerances under different degrees of salt stress. Salt stress causes significant leaf withering and severe membrane injury under a NaCl concentration ≥ 100 mmol/L [6,21]. Salinized soil with a salinity of over 0.3% results in the obvious inhibition of rice plant growth; for example, researchers have observed decreases in plant length, tillers, percentage of filled spikelets, and grain weight [22]. Furthermore, rice is sensitive to salt stress at the booting, heading–flowering, and grain-filling stages; at this stage, salt stress significantly decreases the branch, spikelet, and grain weights, as well as the final grain yield [20,23]. Therefore, it is crucial to explore effective pathways for improving rice yield under salinized soil conditions at the key growth stages.

Appropriate nutrient management is of great importance for growth and yield formation in rice. Among these nutrients, nitrogen plays an important role in the regulation of growth and stress resistance in rice [24]. At the vegetative growth stage, nitrogen fertilizer contributes to increases in plant growth and biomass and improves photosynthesis and tilling [25]. During the reproductive growth stage, nitrogen fertilizer contributes to the promotion of grain filling, increasing grain weight [26]. However, there are different nitrogen fertilizer management methods that have been proposed for use in plants for both normal and salt stress conditions [27]. We previously showed that lower nitrogen levels improved rice growth under salt stress during the reproductive period [10,11,28]. Application of 50% of the normal nitrogen level at the booting and heading stages significantly reduced salinity-induced plant wilting; increased the osmotic adjustment capacity, antioxidant defense capacity, and photosynthesis; and reduced the Na^+^/K^+^ ratio in rice [10,11]. Therefore, planting nitrogen-efficient crop varieties contributes to improving the salt tolerance and yield of the crops under salinized soil conditions, aiding in achieving the goal of reducing fertilizer use and increasing process effectiveness [29,30].

Crops with higher nitrogen-use efficiency are characterized by having a higher tolerance to lower nitrogen levels; they express better growth status and higher nitrogen-use efficiency [31]. Previous studies have shown that nitrogen-efficient rice varieties have better growth characteristics and a better physiological metabolism status in comparison with nitrogen-inefficient rice varieties; this has been shown by their higher biomass, longer roots, higher nitrogen content, higher levels of osmotic adjustment substances, and higher photosynthesis under lower nitrogen levels [32,33]. Furthermore, at a molecular level, many genes and proteins have been shown to have a role in equipping plants with nitrogen efficiency, such as *OsGOGAT1*, *TOND1*, and *OsNPF6.1* [34,35,36]. These physiological characteristics or genes may be supplied for the breeding of higher-nitrogen-use-efficiency crop varieties. However, how do the above traits and genes respond to salt stress? The responses of these traits and genes (in terms of the growth, physiological metabolism, and yield formation) of nitrogen-efficient rice varieties to salt stress remain unknown.

This study aimed to investigate changes in growth, physiological metabolism, gene expression, and yield among rice varieties with different nitrogen-use efficiencies under salt stress conditions. We focused on investigating the effects and their mechanisms by observing the response of nitrogen-efficient rice varieties to salt stress; rice varieties with different nitrogen-use efficiencies were used as the materials. The results of this study provide a theoretical basis and scientific proof for studies on rice production in salinized soil areas.

## 2. Results

### 2.1. Higher N Content and N Metabolism Enzyme Activities in Nitrogen-Efficient Rice Varieties Under Salt Stress Conditions

We analyzed the nitrogen-use-efficiency-related gene expression levels, N content, and N metabolism enzyme activities to evaluate the nitrogen-use efficiency of all the tested rice varieties. *TOND1* is the gene that indicates the tolerance of nitrogen deficiency; overexpression of *TOND1* can increase the tolerance of rice to low nitrogen levels [35]. *OsNPF6.1* encodes a nitrogen transporter protein, which can be used to estimate nitrogen-use efficiency in plants [36]. As shown in Figure 1, salt stress induced the upregulation of *TOND1* and *OsNPF6.1*; meanwhile, the gene expression levels of *TOND1* and *OsNPF6.1* decreased with an increase in salinity. The nitrogen-efficient rice varieties showed higher expression levels of *TOND1* and *OsNPF6.1*. Compared with the CK treatment, the expression levels of *TOND1* were upregulated by 485.5%, 404.7%, and 166.3% under the SS1, SS2, and SS3 treatments, respectively, using nitrogen-efficient rice varieties. Meanwhile, they were upregulated by 88.5% and 21.4% under the SS1 and SS2 treatments, respectively, and decreased by 15.7% under the SS3 treatment using nitrogen-inefficient rice varieties (Figure 1A). Compared with the CK treatment, the expression levels of *OsNPF6.1* were upregulated by 459.2%, 317.4%, and 154.2% under the SS1, SS2, and SS3 treatments, respectively, in the nitrogen-efficient rice varieties. Meanwhile, they were upregulated by 179.1% and 78.6% under the SS1 and SS2 treatments, respectively, and were decreased by 2.2% under the SS3 treatment using nitrogen-inefficient rice varieties (Figure 1B).

Salt stress caused a decrease in nitrate–nitrogen (NO_3_^−^-N); meanwhile, ammonium nitrogen (NH_4_^+^-N) was increased under the SS1 treatment and decreased under the SS2 and SS3 treatments (Figure 1C,D). The activity of nitrate reductase (NR) and NADH-dependent glutamate synthetase (GOCAT) was increased under the SS1 treatment and then decreased with the increase in salinity (Figure 1E,F). The nitrogen-efficient rice varieties accumulated increased NO_3_^−^-N and NH_4_^+^-N contents under the salt stress conditions. Compared with the CK treatment, the NO_3_^−^-N contents of the nitrogen-efficient rice varieties decreased by 15.3%, 31.9%, and 46.6% under the SS1, SS2, and SS3 treatments, respectively; meanwhile, they decreased by 20.4%, 43.6%, and 58.6% in the nitrogen-inefficient rice varieties (Figure 1C). Compared with the CK treatment, the NH_4_^+^-N contents of the nitrogen-efficient rice varieties decreased by 3.2% and 17.1% under the SS2 and SS3 treatments, respectively; meanwhile, they decreased by 11.8% and 24.0% in nitrogen-inefficient rice varieties (Figure 1D). Furthermore, the nitrogen-efficient rice varieties obtained higher activity levels of NR and GOGAT under salt stress conditions (Figure 1E,F). The activity levels of NR in the nitrogen-efficient rice varieties decreased by 14.7% and 41.4% under the SS2 and SS3 treatments, respectively; meanwhile, they were decreased by 32.2% and 53.1% in the nitrogen-inefficient rice varieties (Figure 1E). The activity levels of GOGAT in the nitrogen-efficient rice varieties increased by 27.6% under the SS2 treatment, and decreased by 29.1% under the SS3 treatment; meanwhile, they were decreased by 8.8% and 47.5% in the nitrogen-inefficient rice varieties (Figure 1F).

### 2.2. Seedlings Growth Status of Rice Varieties with Different Nitrogen Efficiencies Under Salt Stress Conditions

As shown in Figure 2, salt stress caused remarkable seedling death in the different rice varieties, as shown by the higher leaf-withering rate and lower chlorophyll and water content. The leaf-withering rate showed an increasing tendency with the increase in salinity, while the chlorophyll and water contents showed decreasing tendencies. Compared with the CK treatment, the leaf-withering rates increased by 28.9–32.1%, 50.3–64.8%, and 77.1–84.9% under the SS1, SS2, and SS3 treatments, respectively, and the chlorophyll contents decreased by 13.3–23.4%, 30.8–45.9%, and 61.7–74.6%, respectively (Figure 2E,F). The average leaf-withering rates of the nitrogen-efficient rice varieties were 30.6%, 52.2%, and 79.6% under the SS1, SS2, and SS3 treatments, respectively, with rates of 29.8%, 62.5%, and 84.3% in the nitrogen-inefficient rice varieties (Figure 2E). The salt stress-induced decrease range in the chlorophyll content of the nitrogen-efficient rice varieties was decreased by 3.8%, 9.7%, and 8.8% under the SS1, SS2, and SS3 treatments, respectively (Figure 2F). Compared with the CK treatment, the average leaf water contents in the nitrogen-efficient rice varieties decreased by 4.0%, 10.5%, and 26.3% under the SS1, SS2, and SS3 treatments, respectively, with decreases of 5.1%, 14.7%, and 35.4% in the nitrogen-inefficient rice varieties (Figure 2G). Compared with the CK treatment, the average root water contents in the nitrogen-efficient rice varieties decreased by 4.3%, 9.1%, and 14.5% under the SS1, SS2, and SS3 treatments, respectively, with decreases of 4.9%, 9.9%, and 18.7% in the nitrogen-inefficient rice varieties (Figure 2H).

### 2.3. Better Root Growth Status of Nitrogen-Efficient Rice Varieties Under Salt Stress Conditions

Salt stress caused an obvious inhibition of root growth, as shown by the decrease of 7.7–56.5% in total root length, 9.1–50.3% in root surface area, 15.1–64.8% in root volume, and 7.3–47.6% in root numbers (Figure 3). Compared with the CK treatment, the average total root length of the nitrogen-efficient rice varieties decreased by 9.8%, 30.2%, and 41.8% under the SS1, SS2, and SS3 treatments, respectively, while they decreased by 20.0%, 38.2%, and 52.4% in the nitrogen-inefficient rice varieties (Figure 3A). The salt stress-induced decreases in the root surface area of the nitrogen-efficient rice varieties of 10.6%, 26.1%, and 38.5% under the SS1, SS2, and SS3 treatments, respectively, while they decreased by 18.6%, 35.9%, and 46.5% in the nitrogen-inefficient rice varieties (Figure 3B). Furthermore, salt stress caused greater decreases in root volume in nitrogen-inefficient rice varieties than in nitrogen-efficient rice varieties (Figure 3C). The salt stress decreased the root number ranges in the nitrogen-efficient rice varieties by 9.0%, 24.6%, and 36.6% under the SS1, SS2, and SS3 treatments, respectively, while they decreased by 18.1%, 32.0%, and 45.2% in the nitrogen-inefficient rice varieties (Figure 3D).

### 2.4. More Accumulation of Osmotic Adjustment Substances and Lower ROS Contents in Nitrogen-Efficient Rice Varieties Under Salt Stress Conditions

Salt stress affected the osmotic regulation system and antioxidant defense system, as shown by an overaccumulation of proline, soluble sugar, O_2_·^−^, and H_2_O_2_, in rice seedlings (Figure 4). Compared with the CK treatment, the proline contents were increased by 63.3–95.7%, 143.7–204.8%, and 109.1–215.7% in leaves and by 30.0–57.1%, 100.2–130.7%, and 64.4–154.1% in roots under the SS1, SS2, and SS3 treatments, respectively, in all tested rice varieties (Figure 4A,B). Compared with the CK treatment, the soluble sugar contents were increased by 63.1–124.0%, 164.8–274.0%, and 144.6–298.2% in the leaves and by 65.9–125.5%, 127.6–223.4%, and 119.0–242.7% in the roots under the SS1, SS2, and SS3 treatments, respectively (Figure 4C,D). The salt stress-induced range increases in proline and soluble sugar contents in both the leaves and the roots in the nitrogen-efficient rice varieties were higher than those in the nitrogen-inefficient rice varieties (Figure 4). With the increase in salinity, the proline and soluble sugar content were increased in the most nitrogen-efficient rice varieties, while they were decreased under the SS3 treatment in the nitrogen-inefficient rice varieties (Figure 4).

Compared with the CK treatment, the O_2_·^−^ contents were increased by 32.8–99.7%, 140.40–277.5%, and 238.1–419.2% in the leaves and by 82.5–191.7%, 222.3–372.0%, and 361.6–517.2% in the roots under the SS1, SS2, and SS3 treatments, respectively (Figure 4E,F). Compared with the CK treatment, the H_2_O_2_ contents were increased by 128.8–146.7%, 339.9–454.1%, and 469.3–729.8% in leaves and by 201.6–270.2%, 402.1–565.9%, and 632.7–835.5% in roots under the SS1, SS2, and SS3 treatments, respectively (Figure 4G,H). The increased ranges for O_2_·^−^ and H_2_O_2_ in the nitrogen-efficient rice varieties were lower than those for the nitrogen-inefficient rice varieties. Compared with the CK treatment, the O_2_·^−^ contents of the nitrogen-efficient rice varieties were increased by 69.1%, 199.1%, and 309.9% in the leaves and by 99.1%, 273.5%, and 405.8% in the roots under the SS1, SS2, and SS3 treatments, respectively; meanwhile, they were increased by 93.4%, 258.7%, and 397.2% in the leaves and by 163.9%, 333.3%, and 479.8% in the roots of the nitrogen-inefficient rice varieties under the SS1, SS2, and SS3 treatments, respectively (Figure 4E,F). The range increases for the salt stress-induced H_2_O_2_ content of the nitrogen-efficient rice varieties decreased by 8.2%, 89.4%, and 190.9% in the leaves in comparison with those of the nitrogen-inefficient rice varieties under the SS1, SS2, and SS3 treatments, respectively; they decreased by 23.1%, 103.1%, and 134.6% in the roots under the SS1, SS2, and SS3 treatments, respectively (Figure 4G,H).

### 2.5. Lower Accumulation of Na^+^ in Nitrogen-Efficient Rice Varieties Under Salt Stress Conditions

Salt stress causes an imbalance in ions and is indicated by a remarkable increase in Na^+^ and a decrease in K^+^ in the leaves and roots of rice varieties with different nitrogen efficiencies (Figure 5). Compared with the CK treatment, the Na^+^ contents were increased by 80.1–124.2%, 225.7–297.6%, and 342.5–501.7% in the leaves and by 115.3–254.5%, 216.0–399.1%, and 351.8–574.1% in the roots under the SS1, SS2, and SS3 treatments, respectively (Figure 5A,B). The K^+^ contents were decreased by 22.7–44.3%, 45.2–66.8%, and 64.0–80.1% in the leaves and 30.6–39.5%, 49.0–65.6%, and 70.2–80.5% in the roots under the SS1, SS2, and SS3 treatments, respectively, compared with the CK treatment (Figure 5C,D). Furthermore, salt stress caused a higher Na^+^/K^+^ ratio in rice seedlings, as shown by the ratio being 142.6–2623.7% higher in the leaves under the SS1–SS3 treatments compared with those under the CK treatment, and 257.0–2938.2% higher in the roots under the SS1–SS3 treatments compared with those under the CK treatment (Figure 5E,F).

Compared with the CK treatment, the Na^+^ contents were increased by 88.4%, 237.6%, and 363.8% in the leaves and by 144.6%, 289.3%, and 427.4% in the roots under the SS1, SS2, and SS3 treatments for the nitrogen-efficient rice varieties; meanwhile, they were increased by 104.3%, 267.1%, and 440.1% in the leaves and by 199.5%, 348.9%, and 524.9% in the roots under the SS1, SS2, and SS3 treatments for the nitrogen-inefficient rice varieties (Figure 5A,B). The salt stress-induced K^+^ decrease range of the nitrogen-efficient rice varieties was lower than that of the nitrogen-inefficient rice varieties. Compared with the CK treatment, the K^+^ contents were decreased by 70.3% in the leaves and 70.9% in the roots of the nitrogen-efficient rice varieties; meanwhile, they were decreased by 77.7% in the leaves and 77.6% in the roots of the nitrogen-inefficient rice varieties under the SS3 treatment (Figure 5C,D). Compared with the CK treatment, the Na^+^/K^+^ ratio was increased by 160.7%, 580.7%, and 1494.1% in the leaves and by 278.8%, 714.9%, and 1714.0% in the roots under the SS1, SS2, and SS3 treatments for the nitrogen-efficient rice varieties; meanwhile, they were increased by 222.0%, 860.0%, and 2328.1% in the leaves and 352.0%, 1105.1%, and 2696.7% in the roots under the SS1, SS2, and SS3 treatments for the nitrogen-inefficient rice varieties (Figure 5E,F).

### 2.6. Higher Expression Levels of Stress-Related Genes in Nitrogen-Efficient Rice Varieties Under Salt Stress Conditions

The ROS-scavenging related genes *OsCATB* and *OsCu/Zn-SOD*, the proline-biosynthesis-related gene *OsP5CS1*, and the Na^+^ and K^+^ transporter protein genes *OsAKT1* and *OsHKT1* were remarkably induced by salt stress; meanwhile, the proline catabolism-related gene *OsPDH1* was suppressed by salt stress (Figure 6). The expression levels of *OsCATB*, *OsCu/Zn-SOD*, *OsP5CS1*, *OsAKT1*, and *OsHKT1* were more induced in the nitrogen-efficient rice varieties, as shown by the higher upregulated range, than they were in the nitrogen-inefficient rice varieties (Figure 6). Compared with the CK treatment, the expression levels of *OsCATB* and *OsCu/Zn-SOD* for the nitrogen-efficient rice varieties were increased by 669.7% and 756.6% under the SS3 treatment; meanwhile, they were increased by 458.0% and 571.8% under the SS3 treatment for the nitrogen-inefficient rice varieties (Figure 6A,B). Compared with the CK treatment, the expression levels of *OsP5CS1* in the nitrogen-efficient rice varieties increased by 662.3%; meanwhile, they were increased by 475.4% in the nitrogen-inefficient rice varieties under the SS3 treatment (Figure 6C). Compared with the CK treatment, the salt-suppressed expression levels of *OsPDH1* in the nitrogen-efficient rice varieties were lower than they were in the nitrogen-inefficient rice varieties, as shown by the following levels: 13.4%, 36.4%, and 66.9% in the nitrogen-efficient rice varieties and 21.6%, 64.0%, and 82.6% in the nitrogen-inefficient rice varieties under the SS1, SS2, and SS3 treatments (Figure 6D). The Na^+^ and K^+^ transporter protein genes *OsAKT1* and *OsHKT1* were more induced in the nitrogen-efficient rice varieties under salt stress conditions. Compared with the CK treatment, the expression levels of *OsAKT1* and *OsHKT1* in the nitrogen-efficient rice varieties were increased by 534.7% and 502.4% under the SS3 treatment; meanwhile, they were increased by 383.9% and 414.4% under the SS3 treatment in the nitrogen-inefficient rice varieties (Figure 6E,F).

### 2.7. Better Growth Status of Nitrogen-Efficient Rice Varieties Under Salinized Paddy Soils

As shown in Figure 7, salt stress caused an obvious inhibition of plant growth for different rice varieties, as shown by the decrease in primary branches (PBs), secondary branches (SBs), panicle length (PL), panicle weight (PW), and filled spikelets (FSs), and the increase in empty spikelets (ESs). The nitrogen-efficient rice varieties showed better growth indexes under salinized paddy soils. Compared with the CK (C) treatment, the PBs of the nitrogen-efficient rice varieties decreased by 1.1%, 5.5%, and 14.5% under the SP1, SP2, and SP3 treatments; meanwhile, they decreased by 4.0%, 16.8%, and 30.6% in the nitrogen-inefficient rice varieties (Figure 7A). Compared with the CK (C) treatment, the SBs of the nitrogen-efficient rice varieties decreased by 3.2%, 14.6%, and 26.4% under the SP1, SP2, and SP3 treatments; meanwhile, they decreased by 8.7%, 21.9%, and 28.7% in the nitrogen-inefficient rice varieties (Figure 7B). There were small differences in the salt stress-induced decreases in PL, PW, and FS between the nitrogen-efficient and -inefficient rice varieties; meanwhile, the salt stress-induced range increases on the ESs of the nitrogen-inefficient rice varieties were higher than those of the nitrogen-efficient rice varieties (Figure 7C–F).

### 2.8. Higher-Yield Component of Nitrogen-Efficient Rice Varieties Under Salinized Paddy Soils

As shown in Figure 8, salt stress caused significant decreases in the yield component indexes of the rice varieties with different nitrogen-use efficiencies. There were non-significant differences in the panicle numbers (PNs) of different rice varieties (Figure 8A). Compared with the CK (C) treatment, the salt stress-induced decrease in spikelets per panicle (SPP) and the percentage of filled spikelets (PFS) showed small differences between different nitrogen efficiency rice varieties (Figure 8B,C). However, the salt stress-induced decreases in the 1000-grain weight (TGW), harvest index (HI), and grain yield (GY) of the nitrogen-inefficient rice varieties were higher than those in the nitrogen-efficient rice varieties (Figure 8D–F). Compared with the CK (C) treatment, the TGW values of the nitrogen-efficient rice varieties decreased by 5.7%, 10.1%, and 13.6% under the SP1, SP2, and SP3 treatments; meanwhile, they decreased by 7.8%, 15.9%, and 19.8% under the SP1, SP2, and SP3 treatments in the nitrogen-inefficient rice varieties (Figure 8D). Compared with the CK (C) treatment, the salt stress-induced decreases in HI for the nitrogen-efficient rice varieties were 3.7%, 1.4%, and 4.9% lower than those for the nitrogen-inefficient rice varieties under the SP1, SP2, and SP3 treatments (Figure 8E). Compared with the CK (C) treatment, the GY of the nitrogen-efficient rice varieties decreased by 12.3%, 29.5%, and 32.9% under the SP1, SP2, and SP3 treatments; meanwhile, they decreased by 15.1%, 32.9%, and 36.9% under the SP1, SP2, and SP3 treatments in the nitrogen-inefficient rice varieties (Figure 8F).

### 2.9. The Way in Which Higher Nitrogen-Use Efficiency Increases Salt Stress Tolerance in Rice

The higher-nitrogen-use-efficiency rice varieties were characterized by a higher efficiency of nitrogen uptake, transport, and use in plants, which contributed to promoting the gene expression network for motivating salt-resistant systems, including osmotic adjustment, antioxidant activity, and ion transport. Further, improvements in plant growth, physiology metabolism, and grain yield were shown in the higher-nitrogen-use-efficiency rice varieties (Figure 9).

## 3. Discussion

Salt stress is characterized by high salinity in soils, which has become an enormous agricultural threat to crop production [37,38]. Rice is moderately sensitive to salt stress. Salt stress causes severe growth inhibition and yield loss for rice, which poses a huge challenge, requiring that we improve rice yields in salinized soil areas in the future [16,39]. Hence, it is of vital importance that we explore effective methods for improving rice yield under salt stress conditions. Appropriate nutrient management strategies contribute to improving rice growth and yield formation in salinized soil [40]. Nitrogen fertilizer plays an important role in promoting plant growth and regulating stress tolerance [41,42]. Previous studies have shown that lower levels of nitrogen application at the reproductive stage can contribute to an improvement in rice growth and yield; this is demonstrated by the alleviation of osmotic stress [43], improvements in the photosynthesis rate [10], and improvements in the osmotic adjustment capacity and antioxidant defense capacity of the plants [11] under salt stress conditions. These results indicate that improvements in nitrogen-use efficiency in rice can potentially cause an improvement in rice’s salt tolerance [40]. In the present study, salt stress caused growth inhibition and yield loss in rice varieties with different nitrogen-use efficiencies. The low-nitrogen-tolerance rice varieties showed higher expression levels of nitrogen-use-efficiency-related genes, higher nitrogen content, and higher nitrogen-use enzyme activity levels (Figure 1), indicating that they are indeed the cause of the higher recorded nitrogen-use efficiency. Under salt stress conditions, the nitrogen-efficient rice varieties showed a higher osmotic adjustment capacity (Figure 4) and K^+^ content (Figure 5), lower ROS content (Figure 4) and Na^+^ content (Figure 5), and higher expression levels for stress-related genes (Figure 6) in rice seedlings, alleviating the problems posed by leaf withering, water shortage (Figure 2), and root injury (Figure 3). Furthermore, the nitrogen-efficient rice varieties showed a better growth status (Figure 7), higher grain weight, and higher yield (Figure 8) under salt paddy soil conditions stimulated at the reproductive growth stage. These results collectively suggest that nitrogen-efficient rice varieties obtain higher nitrogen-use capacities, improving the osmotic adjustment capacity and gene expression levels; then, they can mitigate ion toxicity and oxidative stress, improving the salt tolerance of rice seedlings and ultimately improving grain yield under salt stress conditions.

Rice grown in salinized soil suffers from osmotic stress, ion toxicity, and oxidative stress, which severely inhibit plant growth and the final yield [44,45]. Salt stress-tolerant varieties could adapt to salt stress conditions through an increase in osmotic pressure, a decrease in toxic ion accumulation, and an improvement in ROS-scavenging capacity [46,47]. In the present study, rice varieties with different nitrogen efficiencies showed various phenotypes for salt stress. The growth statuses of the nitrogen-efficient rice varieties were shown to be better than those of the nitrogen-inefficient rice varieties; this was observed through the higher seedling survival (Figure 2), better root growth status (Figure 3), higher levels of osmotic adjustment substances (Figure 4), and lower N^+^ (Figure 5) and ROS levels (Figure 4). Consistently, these traits were associated with higher expression levels for proline biosynthesis, Na^+^-K^+^ transport, and ROS-scavenging genes (Figure 6) in the nitrogen-efficient rice varieties. Furthermore, the fold changes in the above salt stress-related indexes show small differences between the different rice varieties under lower salinity concentrations; this indicates that lower salinity had little effect on rice seedlings, which can be said for the nitrogen content as well (Figure 1). However, with the increase in salinity, the fold changes (increasing or decreasing) in the above salt stress-related indexes in the nitrogen-efficient rice varieties were remarkably lower than those of the nitrogen-inefficient rice varieties. This finding suggests that the nitrogen-efficient rice varieties could better maintain the stability of their physiological metabolism, allowing them to adapt to salt stress conditions more effectively. These results collectively suggest that nitrogen-efficient rice varieties could obtain higher nitrogen levels and transport efficiencies, allowing them to maintain stable physiological metabolism processes to cope with salt stress; this finding is in accordance with those of previous studies [48,49].

Higher nitrogen-use efficiency under lower-nitrogen-application conditions is a vital characteristic for nitrogen-efficient plant varieties to have [24,50]. Previous studies have shown that under lower-nitrogen conditions, nitrogen-efficient rice varieties obtain more biomass, greater root length, higher photosynthesis rates, and higher nitrate reductase activities than nitrogen-inefficient rice varieties [32,51]. Qi et al. [33] reported that nitrogen-efficient rice varieties obtain higher dry weight, net photosynthetic rate, and antioxidant defense capacity under conditions with lower nitrogen supply levels. In the present study, salt stress induced the expression of nitrogen-use-efficiency-related genes (Figure 1), the accumulation of nitrogen, and an increase in the activities of nitrogen-use enzymes (Figure 1), indicating that nitrogen was indeed activated in response to salt stress. This result implies that nitrogen plays a role in the defense against salt stress in rice. However, the levels of nitrogen content activation and enzyme activities decreased with the increase in salinity. Nevertheless, the gene expression levels, nitrogen content, and enzyme activities of the nitrogen-efficient rice varieties were remarkably higher than those of the nitrogen-inefficient rice varieties under salt stress; this finding indicates that nitrogen-efficient rice varieties could better stabilize the nitrogen balance and transport efficiency, enabling the plant to maintain a normal physiological metabolism process.

Plants respond to environmental stresses by reprogramming their gene expression networks; this is an important mechanism for plants as it enables them to cope with stresses [52]. In this study, salt stress induced the expression of proline biosynthesis, ion transport, and ROS-scavenging genes, indicating that salt stress caused osmotic stress, ion toxicity, and oxidative stress to rice seedlings (Figure 6). However, the expression levels were decreased by higher salinity, indicating that these physiological metabolism processes were damaged by higher-level salt stress. Nevertheless, higher gene expression levels were observed for the nitrogen-efficient rice varieties compared with the nitrogen-inefficient rice varieties, with an increase in salinity (Figure 6). These results suggest that higher nitrogen levels contribute to effectively improving gene expression networks, enabling them to cope with salt stress in nitrogen-efficient rice varieties.

As shown in Figure 9, rice varieties with higher nitrogen-use efficiency are characterized by higher activity levels among N-use-related enzymes (Figure 1); accordingly, they can accumulate more nitrogen (Figure 1) in the plants. This is in accordance with the findings of previous studies [32,33]. Furthermore, the gene expression levels of N-use efficiency, osmoregulation, ion transport, and antioxidant defense were more upregulated in the nitrogen-use-efficient rice varieties (Figure 1 and Figure 6), which promoted a better balance in osmotic adjustment, ROS accumulation, and Na-K transport. Collectively, these results indicate that plants with higher nitrogen-use efficiency may have a better capacity for taking up, transporting, and using nitrogen for improving growth, physiological metabolism, and yield formation under salt stress conditions. It would be interesting to investigate the molecular mechanism of how nitrogen signals effect plant growth, physiological metabolism, and stress tolerance. Therefore, it is conceivable that strengthening N-use efficiency may help in improving stress tolerance in rice and could be used as a breeding strategy for producing resistant rice varieties.

## 4. Materials and Methods

### 4.1. Plant Materials

Eight japonica rice varieties with different nitrogen-use efficiencies were provided by the Rice Research Institute of Jilin Agricultural University. These were used as the materials in this study. Dongdao-4 (D4), Changbai-9 (C9), Jinongda-667 (J-667), and Jinongda-668 (J-668) were used as the nitrogen-efficient rice (NER) varieties with higher N-use efficiency under low-nitrogen conditions; their periods of duration were 131 days, 137 days, 137 days, and 144 days, respectively, according to the description of China Rice Data Center. Nipponbare (NB), Jijing-88 (J-88), Jiyujing (JYJ), and Jinongda-669 (J-669) were used as the nitrogen-inefficient rice (NIR) varieties, with lower N-use efficiency under low-nitrogen conditions; their periods of duration were 141 days, 148 days, 135 days, and 137 days, respectively, according to the description of China Rice Data Center. The nitrogen-use characteristics of these rice varieties were identified according to the method presented in a previous study [33]. Under low-nitrogen conditions, the rice materials were treated with low nitrogen levels, stimulated by 1/4 N at the seedling stage; various growth and nitrogen-utilization indexes were measured to calculate the low-nitrogen-tolerance comprehensive score value (D) of each rice material [33]. The comprehensive score values (D) of D4, C9, -J667, J-668, NB, J-88, JYJ, and J-669 were 0.685, 0.663, 0.582, 0.566, 0.486, 0.499, 0.426, and 0.455, respectively, under low-nitrogen conditions.

### 4.2. Salt Stress Application at the Seedling Stage

Seeds from the different rice varieties were cultivated to approximately the three-leaf stage in a controlled growth chamber under 28 °C day/22 °C night, a 12 h photoperiod, and a 350 mmol photon m^−2^ s^−1^ light intensity; this was conducted according to the experimental methods described in our previous study [52]. The rice seedlings were then transferred to distilled water (CK) or salt-stressed (SS) conditions. Salt stresses of 50, 100, and 150 mmol/L NaCl were applied to the rice seedlings; these conditions are represented by SS1, SS2, and SS3, respectively. All treatments were conducted in a controlled growth chamber with three biological repeats under the following conditions: 28 °C day/22 °C night, 12 h photoperiod, and 350 mmol photon m^−2^ s^−1^ light intensity. After seven days of salt stress, the rice leaves and roots were sampled to measure growth, physiological indices, and gene expression levels.

### 4.3. Salt Stress Application at the Reproductive Growth Stage

Seeds from the different rice varieties were sown in a nursery bed with humus soil, as described by Liu et al. [53]. Healthy rice seedlings from the different varieties at approximately the three-leaf stage were transplanted into pots, with a height of 28 cm and an internal diameter of 25 cm, containing 10 kg of experimental paddy soil for further growth. The paddy soil was pounded, sifted, dried, and then mixed before being placed into the pots; the soil contained 26.3 g/kg organic matter, 1.9 g/kg total nitrogen, 15.3 mg/kg readily available phosphorus, and 106.8 mg/kg readily available potassium. A previously published fertilization method was followed [53]. Briefly, 4.5 g of a compound fertilizer, in which the levels of N, P, and K were all 15%, was applied to each experimental paddy soil pot for the base fertilizer, and 3.0 g of urea was applied to each pot at the tillering stage for the after manuring. Two clumps of three rice seedlings were transplanted to each pot. Rice plants were cultivated to the heading stage under normal conditions using traditional cultivation management methods. Then, the rice plants of different varieties were treated with salt stress-stimulated conditions using salinity levels of 0.2%, 0.4%, and 0.6% at the day of full-heading stage, represented by SP1, SP2, and SP3, respectively. Different weights of pure analysis-grade NaCl, according to the soil weight per pot, were applied to the soils, and the unstressed soil was set as the control (C). All treatments were conducted under normal conditions with three biological repeats using traditional cultivation management methods. At the mature stage, the rice plants were sampled to measure the growth indexes and yield components.

### 4.4. Measurement of Seedling Growth

Photographs of the growing rice seedlings were taken on the 7th day of the treatments. The leaf-withering rate was investigated and recorded as 1 if the whole leaf was dry and brown, whereas it was recorded as 0.5 if half of the leaf was dry and brown [52]. According to the description by Liu et al. [14], five rice seedlings were randomly selected in each treatment group to measure the root growth status. These seedlings were scanned with an Epson Expression 10000XL (Epson America Inc., Long Beach, CA, USA) on the 7th day of salt stress. The resulting images were digitized with *WinRHIZO* (Regent Instruments Canada Inc., Ville de Québec, QC, Canada), and the total root length (TRL), total root surface area (RSA), total root volume (RV), and root numbers (RNs) were recorded.

### 4.5. Measurement of the Chlorophyll Content of Rice Seedlings

Chlorophyll content was measured as described by Wellburn and Lichtenthaler [54], with some modifications as described by Liu et al. [52]. Briefly, leaf samples (0.1 g) were extracted using a 5 mL mixture of ethanol (2.5 mL) and acetone (2.5 mL). The absorbance of the supernatant was measured at 645 and 663 nm using a spectrophotometer (UV-2700; Shimadzu, Kyoto, Japan). The total chlorophyll content was calculated using the following formula: (20.29 × A_645_ + 8.05 × A_663_)V/(1000 × W). The “V” and “W” indicate the volume of the extracting solution and the weight of leaf samples, respectively, in the formula.

### 4.6. Measurement of the Relative Water Content of Rice Seedlings

The relative water content of leaves and roots were measured according to the method described by Wei et al. [55]. Briefly, five rice seedlings from each treatment were cut, divided into shoots and roots, and the fresh weights of the shoots and roots were measured. Subsequently, the samples were dried in a forced air oven at 105 °C for 2 h and then heated at 70 °C until a stable mass was reached before the dry mass was determined. The relative water content (%) was calculated based on the fresh and dry weights.

### 4.7. Measurement of Plant Growth, Grain Yield, and Yield Components at the Mature Stage

At the mature stage, all rice plants were harvested to determine the following parameters: shoot length (SL); shoot dry weight (SDW); primary branches (PBs) and secondary branches (SBs) of panicles; panicle length (PL); panicle weight (PW); yield components, including panicle number (PN), spikelets per panicle (SP), number and weight of filled spikelets (FSs) and empty spikelets (ESs), percentage of filled spikelets (PFS), 1000-grain weight (TGW), harvest index (HI), and grain yield (GY). The GY and yield components were measured according to the method described by Liu et al. [53]. The TGW and aboveground biomass were adjusted to a 0.14 g/g moisture content on a dry weight basis. HI was calculated as the GY divided by the aboveground biomass.

### 4.8. Measurement of Proline and Soluble Sugar Content

A measure of 0.1 g of dried rice leaves and roots, combined with 10 mL deionized water, was placed into a centrifuge tube. After centrifuge and boiling, the sample was used for the measurement of proline and soluble sugars in the leaves and roots. Proline contents were measured by the sulfosalicylic acid method, and soluble sugars were detected with anthrone colorimetry, according to the description by Liu et al. [11].

### 4.9. Measurement of Na^+^ and K^+^ Content

The sodium (Na^+^) and potassium (K^+^) contents in the rice leaves and roots were measured according to Liu et al. [11]. The dried leaf or root samples (0.1 g) were digested completely with the mixture of HNO_3_ and HClO_4_ (*v*/*v* = 2:1), and then the solution was diluted to 50 mL with deionized water. The Na^+^ and K^+^ concentrations in the solution were determined using flame emission spectrometry (FP6410, Shanghai precision and scientific instrument Co., Ltd., Shanghai, China).

### 4.10. Measurement of ROS Levels

The O_2_·^−^ content was measured by monitoring nitrite formation from hydroxylamine in the presence of O_2_·^−^, using the method described by Elstner and Heupel [56]. The absorbance of the aqueous solution at 530 nm was measured to calculate the levels of O_2_·^−^ from the chemical reaction of O_2_·^−^ and hydroxylamine. The H_2_O_2_ content was measured by monitoring the absorbance at 415 nm of the titanium–peroxide complex, as previously described by Brennan and Frenkel [57]. The absorbance values in the aqueous solution were measured at 415 nm to calculate the levels of H_2_O_2_. According to the theory and methods aforementioned, the analytical reagents used to measure the H_2_O_2_ and O_2_·^−^ contents were acquired from the determination kit according to the manufacturer’s instructions (Suzhou Michy Biomedical Technology Co., Ltd., Suzhou, China) [7,53].

### 4.11. Measurement of Nitrate–Nitrogen and Ammonium Nitrogen Contents in Rice

The rice samples (0.2 g) were ground to a powder and extracted in 10 mL of distilled water for 2.5 h, and the supernatant was reserved for further measurement. The nitrate–nitrogen (NO_3_^−^-N) content of the solution was determined after mixing 0.2 mL of solution with 10% (*w*/*v*) salicylic acid in 96% sulfuric acid using a spectrophotometer under 410 nm [58]. The values were quantified after generating a standard curve as described by Cataldo et al. [59]. To assess the content of ammonium nitrogen (NH_4_^+^-N), the supernatant solution was mixed with phenol. After shaking off, sodium hypochlorite was added in the solution and then maintained at 25 °C for 1 h. The absorbance of reaction mixture was measured under 625 nm using a spectrophotometer (UV-2700; Shimadzu, Kyoto, Japan). The ammonium nitrogen content was proportional to the absorption value [58].

### 4.12. Measurement of N-Metabolism-Related Enzyme Activities

Nitrate reductase (NR, EC 1.7.1.3) has been proposed as an important index of nitrate incorporation in plants [60]. According to the description in a previous study [58], the NR activity level was measured by the reduction rate of NADH in the reaction of the reduction of nitrate to nitrite, which was catalyzed by NR. The rice samples (0.5 g) were homogenized in 3 mL of ice-cold extraction buffer consisting of 100 mM HEPES-KOH (pH: 7.5), 1% (*w*/*v*) polyvinylpolypyrrolidone, 1 mM EDTA, 5 mM Mg(CH_3_COOH)_2_, 5 μM Na_2_MoO_4_, 10% (*v*/*v*) glycerol, 5 mM dithiothreitol (DTT), 0.1% Triton X-100, 1% (*w*/*v*) BSA, 20 μM flavin adenine dinucleotide, 25 μM leupeptin, and 0.5 mM phenylmethylsulfonyl fluoride, the latter with in 99% ethanol. Then, 100 μL of the extraction mixture was transferred to an Eppendorf tube with 0.5 mL of assay medium containing 100 mM HEPES-KOH (pH 7.5), 5 mM EDTA, 5 mM KNO_3_, and 0.25 mM NADH added in it. The mixture was incubated at 25 °C for 30 min and then the reaction was stopped by adding a stop reagent containing 75 μL of 0.15 mM phenazine methosulphate and 25 μL of 0.6 M Zn(CH_3_COOH)_2_. The samples were incubated at room temperature for 15 min and 300 μL of a color reagent containing 0.02% N-(1-naphthyl) ethylenediamine dihydrochloride and 1% sulfanilamide in 3 M HCl was added into it. After centrifugation, the supernatants were measured at 540 nm to measure the activity of NR. The catalytic reduction of 1 nmol NADH per minute per g of fresh weight sample was represented as one unit of NR activity.

NADH-dependent glutamate synthetase (GOGAT, EC 1.4.1.14) was used to catalyze the formation of the glutamic acid through the transfer of glutamine to alpha-ketoglutaric acid [61]. The rice samples (0.5 g) were homogenized in 10 mM Tris-HCl buffer (pH: 7.6) with 1 mM β-mercaptoethanol, 1 mM EDTA, and 1 mM MgCl_2_ and then centrifuged. The absorbance of the reaction mixture was measured at 340 nm after adding a mixed liquor containing 0.875 mL of 25 mM Tris-HCl buffer (pH: 7.6), 0.1 mL of 3 mM NADH, 0.2 mL of 20 mM l-glutamine, 50 μL of 10 mM KCl, 25 μL of 0.1 M 2-oxoglutarate, and 0.25 mL of enzyme extract. One unit of the GOGAT activity was expressed by the consumption of 1 nmol of NADH per g of tissue per minute, according to the previous study [58].

### 4.13. RNA Isolation and Quantitative Real-Time PCR (qRT-PCR)

The rice leaves were sampled in liquid nitrogen and ground using a bench-top ball-mill (Scientz-48, Ningbo Scientz Biotechnology Co. Ltd., Ningbo, China) at 50 Hz for 30 s. Total RNA was extracted with TRIzol reagent (TaKaRa Bio Tokyo, Japan) and first-strand cDNA was synthesized using M-MLV reverse transcriptase (Thermo, Carlsbad, CA, USA), according to the manufacturer’s protocols. Quantitative real-time PCR (qRT-PCR) was performed to determine the transcriptional expression of genes, including two nitrogen-use-efficiency-related genes, *TOND1* [35] and *OsNPF6.1* [36], and four stress-related genes: *OsP5CS1* and *OsPDH1* [11], *OsCu/Zn-SOD* and *OsCATB* [52,62], and *OsAKT1* and *OsHKT1* [62]. Gene-specific primers were designed using Primer 5.0 software. The gene-specific primer pairs used for qRT-PCR are shown in Table 1.

The housekeeping gene *β-actin* (GenBank ID: X15865.1) was used as an internal gene. PCR was conducted in a 20 µL reaction mixture containing 1.6 µL of cDNA template (50 ng), 0.4 µL of 10 mM specific forward primer, 0.4 µL of 10 mM specific reverse primer, 10 µL of 2×SYBR^®^
*Premix Ex Taq*^TM^ (TaKaRa, Bio Inc., Kusatsu-Shi, Japan), and 7.6 µL of double-distilled H_2_O in a PCR machine (qTOWER2.2. Analytic Jena GmbH. Jena, GER, Jena, Germany). The procedure was performed as follows: 1 cycle for 30 s at 95 °C, 40 cycles for 5 s at 95 °C, and 20 s at 60 °C, and 1 cycle for 60 s at 95 °C, 30 s at 55 °C, and 30 s at 95 °C for the melting curve analysis. The level of relative expression was computed using the 2^−△△CT^ method [63].

### 4.14. Experimental Design and Statistical Analyses

All the experiments were performed with three biological replicates. Statistical analyses were performed using the statistical software SPSS 21.0 (IBM Corp., Armonk, NY, USA). Based on a one-way analysis of variance (ANOVA), Duncan’s multiple range test (DMRT) was used to compare differences in the means among treatments. The significance level was *p* < 0.05.

## 5. Conclusions

In summary, in this study, nitrogen-efficient rice varieties achieved the following outcomes: they obtained higher nitrogen-use efficiency; they maintained higher nitrogen levels under salt stress conditions, enabling them to motivate the gene expression regulatory network and increase the capacity of osmotic adjustment, ion transport, and antioxidant defense; and they showed alleviated osmotic stress, ion toxicity, and oxidative stress; in turn, they showed improved plant growth, physiological metabolism status, and grain yield under salt stress conditions.

## Figures and Tables

**Figure 1 plants-14-00556-f001:**
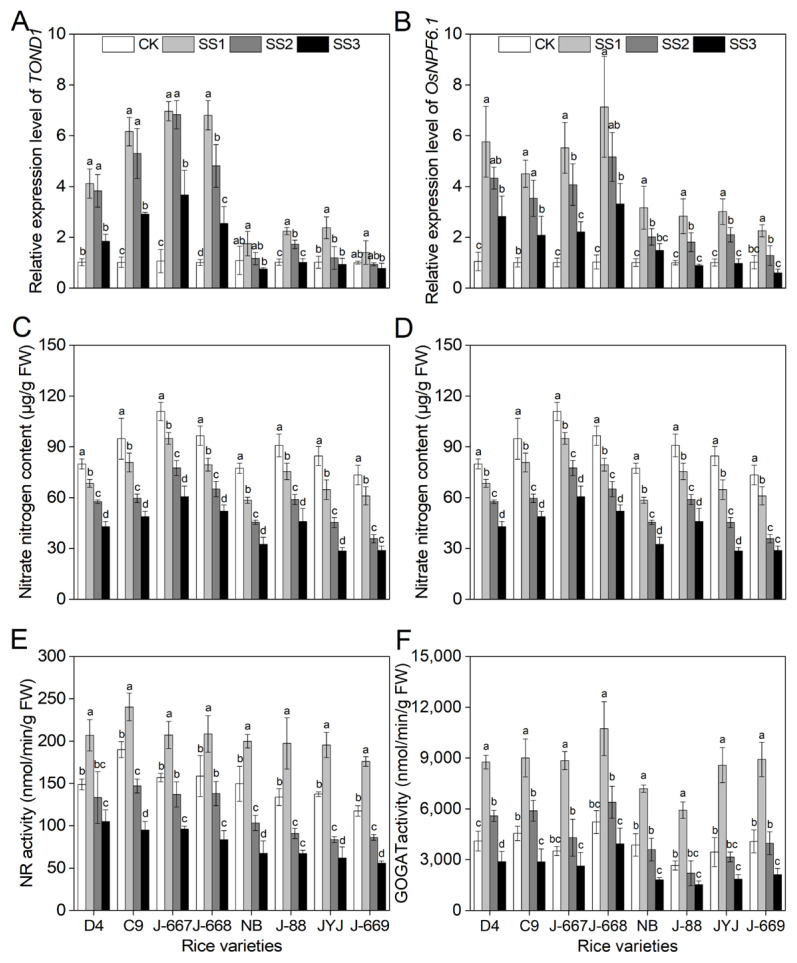
Expression levels of nitrogen-use genes, nitrogen content, and nitrogen utilization-related enzyme activities of different nitrogen-efficient rice varieties under salt stress conditions. Dongdao-4 (D4), Changbai-9 (C9), Jinongda-667 (J-667), and Jinongda-668 (J-668) were used as the nitrogen-efficient rice (NER) varieties. Nipponbare (NB), Jijing-88 (J-88), Jiyujing (JYJ), and Jinongda-669 (J-669) were used as the nitrogen-inefficient rice (NIR) varieties. Two-week-old rice seedlings were treated with distilled water (CK) and salt stress-stimulated using 50 (SS1), 100 (SS2), and 150 (SS3) mmol/L NaCl. The expression levels of nitrogen-use-related genes *TOND1* (**A**) and *OsNPF6.1* (**B**) were measured after 72 h of salt stress treatments. A quantitative real-time polymerase chain reaction was performed using *OsACT1* as an internal standard. The expression level of distilled water (CK) was set as the unit to calculate the expression levels in each rice variety, shown as fold changes relative to the CK. The nitrate–nitrogen (NO_3_^−^-N) content (**C**), ammonium nitrogen (NH_4_^+^-N) content (**D**), and activity levels of nitrate reductase (NR) (**E**) and NADH-dependent glutamate synthetase (GOGAT) (**F**) were measured on the 7th day of salt stress treatments. Values are presented as means ± SD, *n = 3*. Different letters on the column represent significant differences (*p* < 0.05) between different treatments of the same rice varieties based on Duncan’s test.

**Figure 2 plants-14-00556-f002:**
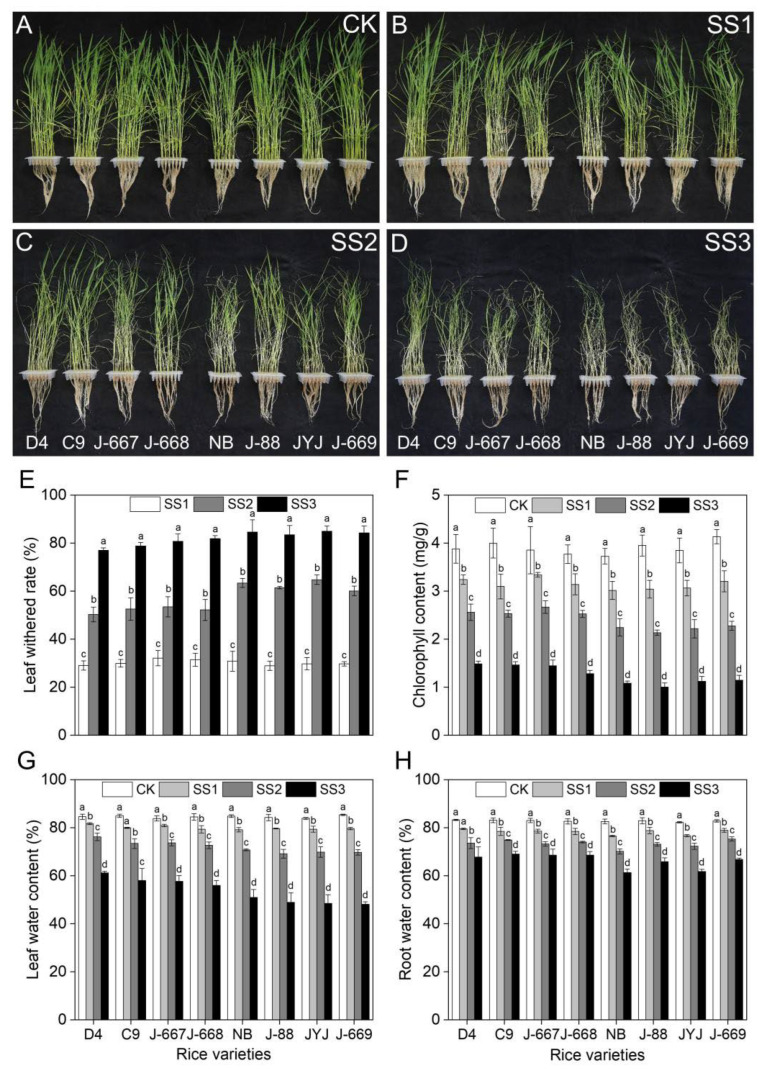
Seedlings growth status of different nitrogen-efficient rice varieties under salt stress conditions. Dongdao-4 (D4), Changbai-9 (C9), Jinongda-667 (J-667), and Jinongda-668 (J-668) were used as the nitrogen-efficient rice (NER) varieties. Nipponbare (NB), Jijing-88 (J-88), Jiyujing (JYJ), and Jinongda-669 (J-669) were used as the nitrogen-inefficient rice (NIR) varieties. Two-week-old rice seedlings were treated with distilled water (CK) and salt stress-stimulated using 50 (SS1), 100 (SS2), and 150 (SS3) mmol/L NaCl. Photographs of seedling growth under the CK (**A**), SS1 (**B**), SS2 (**C**), and SS3 (**D**) treatments were taken on the 7th day of the salt stress treatments. The leaf-withering rate (**E**), chlorophyll content (**F**), leaf water content (**G**), and root water content (**H**) of rice seedlings was counted on the 7th day of salt stress treatments. Values are presented as means ± SD, *n* = 3. Different letters on the column represent significant differences (*p* < 0.05) between different treatments of the same rice varieties based on Duncan’s test.

**Figure 3 plants-14-00556-f003:**
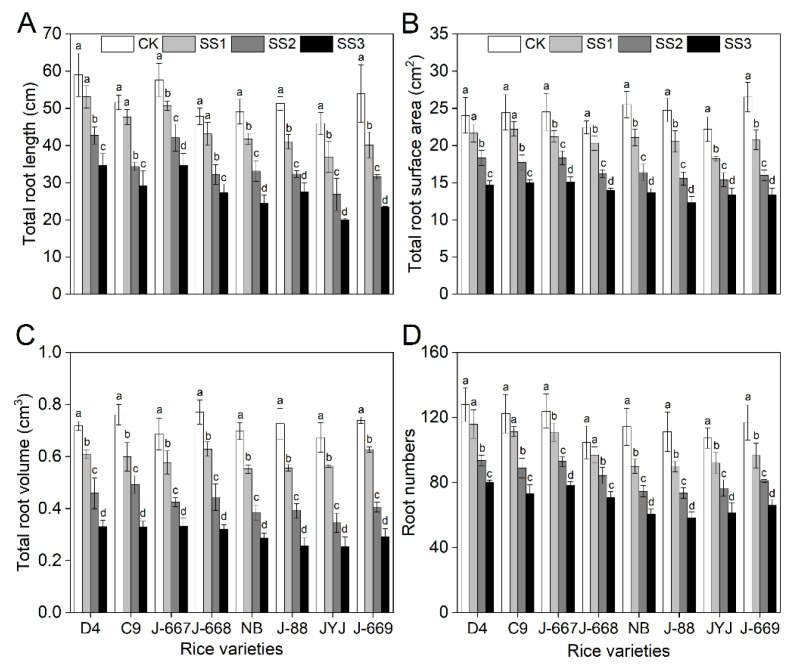
Root growth indexes of different nitrogen-efficient rice varieties under salt stress conditions. Dongdao-4 (D4), Changbai-9 (C9), Jinongda-667 (J-667), and Jinongda-668 (J-668) were used as the nitrogen-efficient rice (NER) varieties. Nipponbare (NB), Jijing-88 (J-88), Jiyujing (JYJ), and Jinongda-669 (J-669) were used as the nitrogen-inefficient rice (NIR) varieties. Two-week-old rice seedlings were treated with distilled water (CK) and salt stress-stimulated using 50 (SS1), 100 (SS2), and 150 (SS3) mmol/L NaCl. Total root length (**A**), total root surface areas (**B**), total root volume (**C**), and root numbers (**D**) were measured on the 7th day of salt stress treatments. Values are presented as means ± SD, *n* = 3. Different letters on the column represent significant differences (*p* < 0.05) between different treatments of the same rice varieties based on Duncan’s test.

**Figure 4 plants-14-00556-f004:**
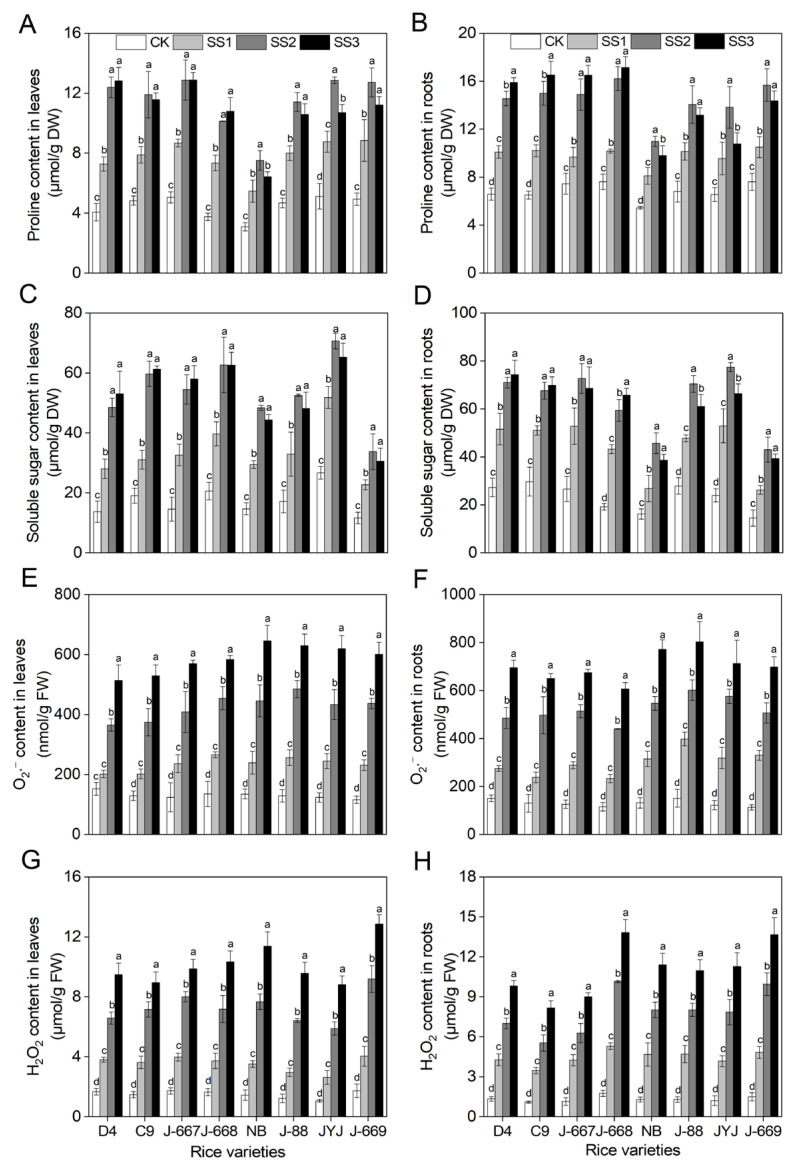
Osmotic adjustment substances and reactive oxygen species (ROSs) contents of different nitrogen-efficient rice varieties under salt stress conditions. Dongdao-4 (D4), Changbai-9 (C9), Jinongda-667 (J-667), and Jinongda-668 (J-668) were used as the nitrogen-efficient rice (NER) varieties. Nipponbare (NB), Jijing-88 (J-88), Jiyujing (JYJ), and Jinongda-669 (J-669) were used as the nitrogen-inefficient rice (NIR) varieties. Two-week-old rice seedlings were treated with distilled water (CK) and salt stress-stimulated using 50 (SS1), 100 (SS2), and 150 (SS3) mmol/L NaCl. Proline content (**A**,**B**), soluble sugar content (**C**,**D**), O_2_·^−^ content (**E**,**F**), and H_2_O_2_ (**G**,**H**) were measured on the 7th day of salt stress treatments. Values are presented as means ± SD, *n* = 3. Different letters on the column represent significant differences (*p* < 0.05) between different treatments of the same rice varieties based on Duncan’s test.

**Figure 5 plants-14-00556-f005:**
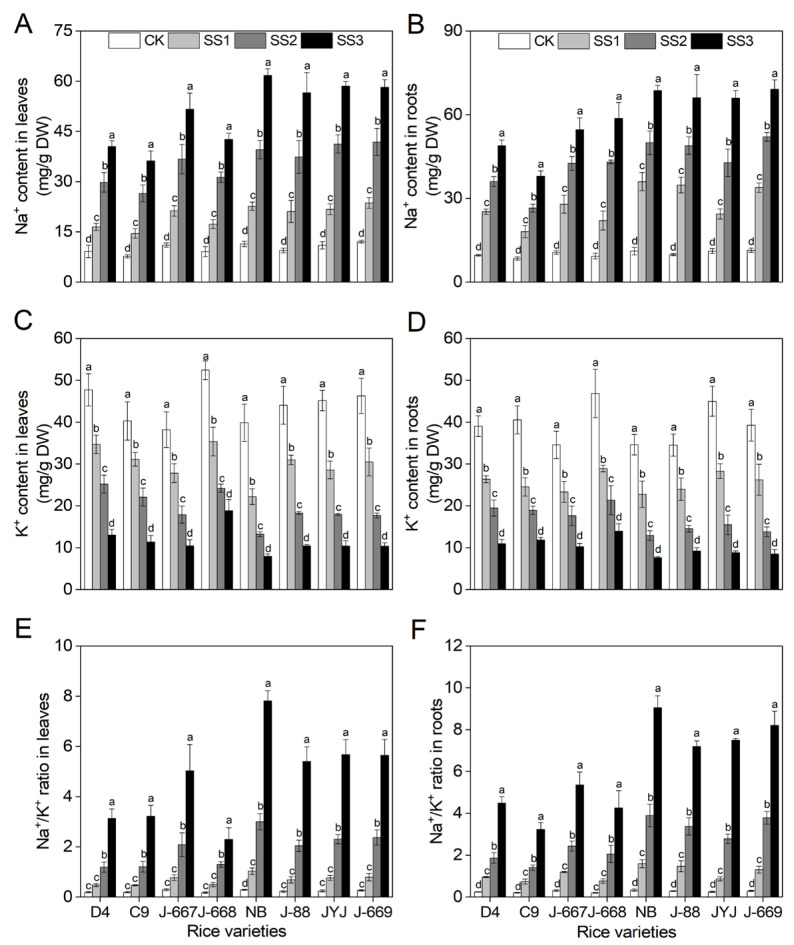
Na^+^ and K^+^ contents of different nitrogen-efficient rice varieties under salt stress. Dongdao-4 (D4), Changbai-9 (C9), Jinongda-667 (J-667), and Jinongda-668 (J-668) were used as the nitrogen-efficient rice (NER) varieties. Nipponbare (NB), Jijing-88 (J-88), Jiyujing (JYJ), and Jinongda-669 (J-669) were used as the nitrogen-inefficient rice (NIR) varieties. Two-week-old rice seedlings were treated with distilled water (CK) and were salt stress-stimulated using 50 (SS1), 100 (SS2), and 150 (SS3) mmol/L NaCl. Na^+^ content (**A**,**B**), K^+^ content (**C**,**D**), and Na^+^/K^+^ ratio (**E**,**F**) were measured on the 7th day of salt stress treatments. Values are presented as means ± SD, *n* = 3. Different letters on the column represent significant differences (*p* < 0.05) between different treatments of the same rice varieties based on Duncan’s test.

**Figure 6 plants-14-00556-f006:**
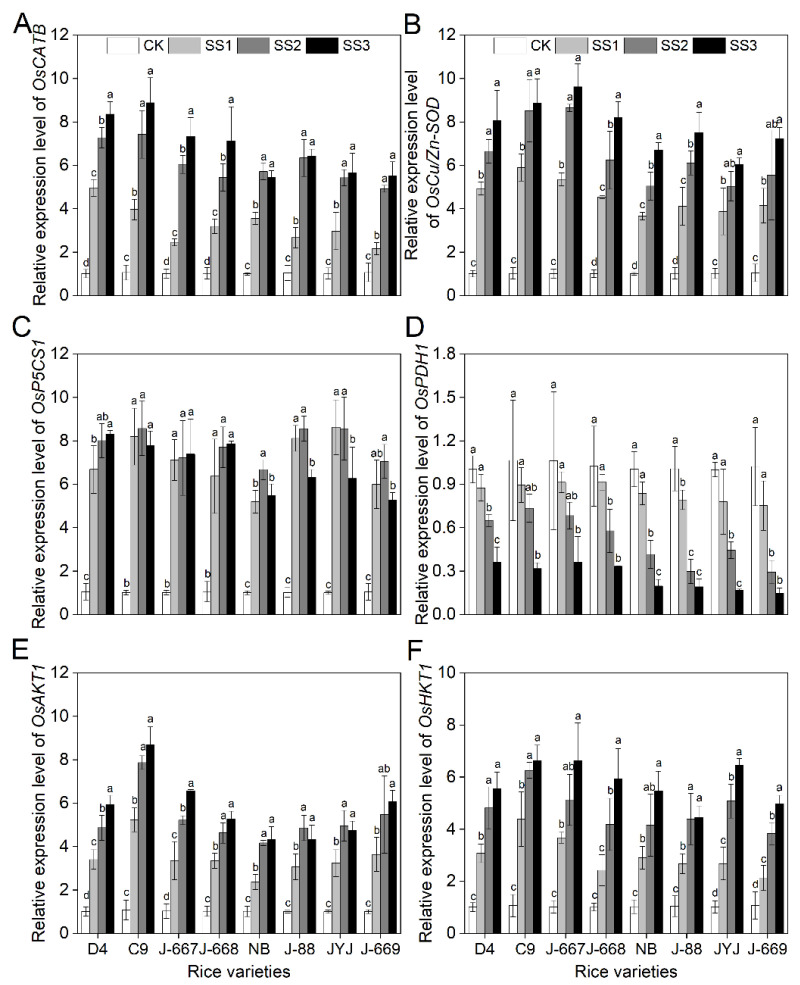
Expression levels of stress-related genes of different nitrogen-efficient rice varieties under salt stress conditions. Dongdao-4 (D4), Changbai-9 (C9), Jinongda-667 (J-667), and Jinongda-668 (J-668) were used as the nitrogen-efficient rice (NER) varieties. Nipponbare (NB), Jijing-88 (J-88), Jiyujing (JYJ), and Jinongda-669 (J-669) were used as the nitrogen-inefficient rice (NIR) varieties. Two-week-old rice seedlings were treated with distilled water (CK) and salt stress-stimulated using 50 (SS1), 100 (SS2), and 150 (SS3) mmol/L NaCl. Expression levels of stress-related genes *OsCATB* (**A**), *OsCu/Zn-SOD* (**B**), *OsP5CS1* (**C**), *OsPDH1* (**D**), *OsAKT1* (**E**), and *OsHKT1* (**F**) were measured at 72 h of salt stress treatments. A quantitative real-time polymerase chain reaction was performed, using *OsACT1* as an internal standard. The expression levels of distilled water (CK) were set as the unit to calculate the expression levels in each rice variety, shown as fold changes relative to the CK. Values are presented as means ± SD, *n* = 3. Different letters on the column represent significant differences (*p* < 0.05) between different treatments of the same rice varieties based on Duncan’s test.

**Figure 7 plants-14-00556-f007:**
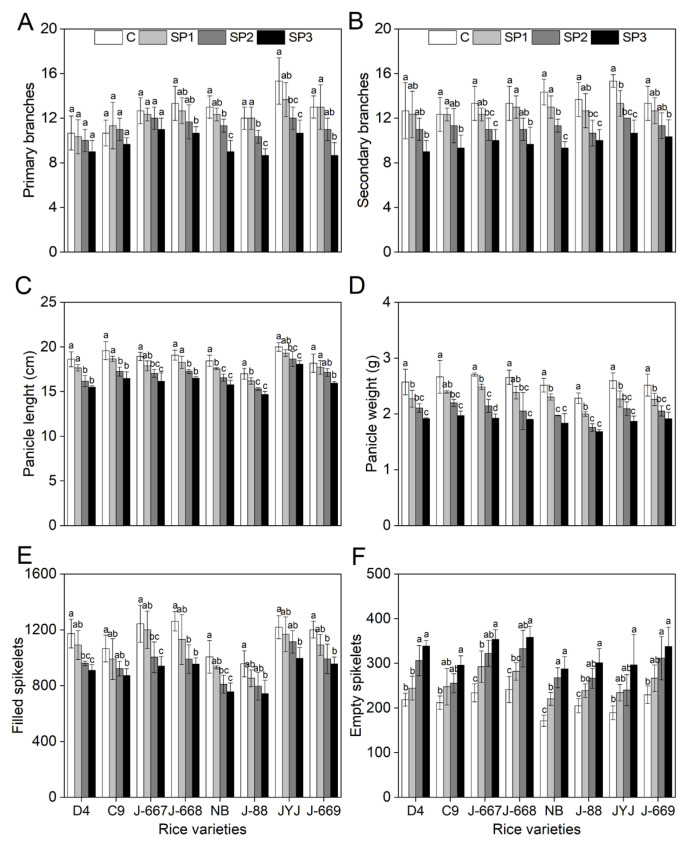
Growth indexes of different nitrogen-efficient rice varieties under salt paddy soil conditions. Dongdao-4 (D4), Changbai-9 (C9), Jinongda-667 (J-667), and Jinongda-668 (J-668) were used as the nitrogen-efficient rice (NER) varieties. Nipponbare (NB), Jijing-88 (J-88), Jiyujing (JYJ), and Jinongda-669 (J-669) were used as the nitrogen-inefficient rice (NIR) varieties. The rice plants were planted in unstressed soil (C) or salt paddy soil (SP) conditions stimulated by salinities of 0.2 (SP1), 0.4% (SP2), and 0.6% (SP3) from the full-heading stage. Primary branches (**A**), secondary branches (**B**), panicle length (**C**), panicle weight (**D**), filled spikelets (**E**), and empty spikelets (**F**) were measured at the mature stage. Values are presented as means ± SD, *n* = 3. Different letters on the column represent significant differences (*p* < 0.05) between different treatments of the same rice varieties based on Duncan’s test.

**Figure 8 plants-14-00556-f008:**
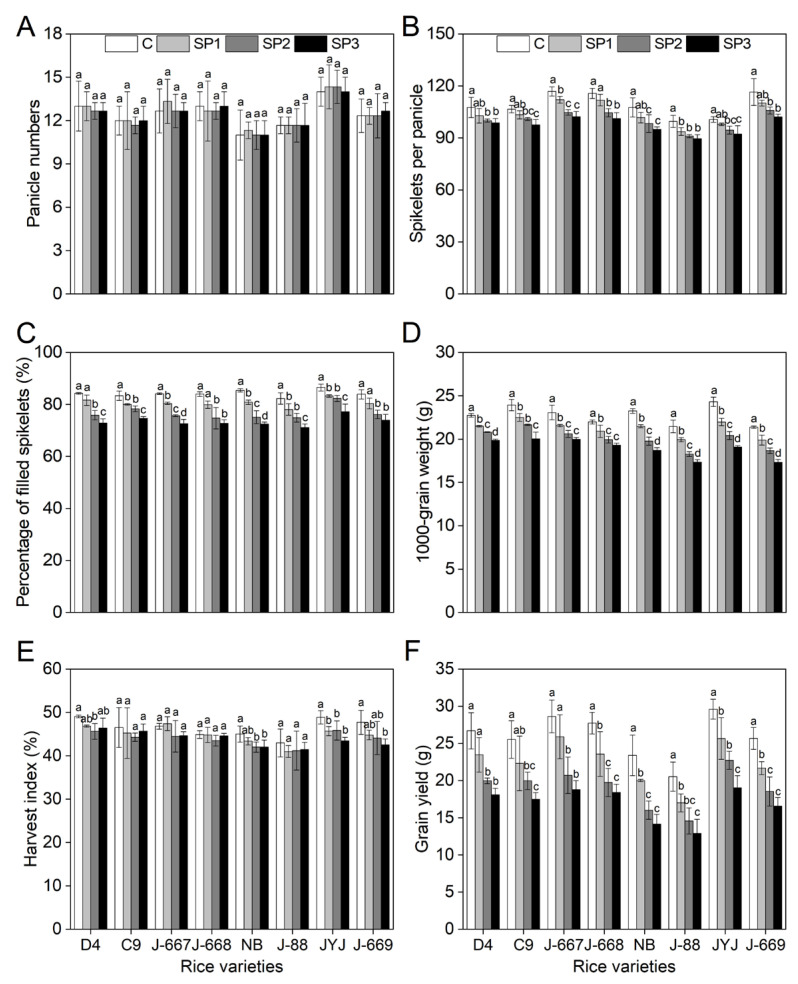
Yield component indexes of different nitrogen-efficient rice varieties under salt paddy soil conditions. Dongdao-4 (D4), Changbai-9 (C9), Jinongda-667 (J-667), and Jinongda-668 (J-668) were used as the nitrogen-efficient rice (NER) varieties. Nipponbare (NB), Jijing-88 (J-88), Jiyujing (JYJ), and Jinongda-669 (J-669) were used as the nitrogen-inefficient rice (NIR) varieties. The rice plants were planted in unstressed soil (C) or salt paddy soil (SP) conditions, stimulated by salinities of 0.2 (SP1), 0.4% (SP2), and 0.6% (SP3) from the full-heading stage. Panicle numbers (**A**), spikelets per panicle (**B**), percentage of filled spikelets (**C**), 1000-grain weight (**D**), harvest index (**E**), and grain yield (**F**) were measured at the mature stage. Values are presented as means ± SD, *n* = 3. Different letters on the column represent significant differences (*p* < 0.05) between different treatments of the same rice varieties based on Duncan’s test.

**Figure 9 plants-14-00556-f009:**
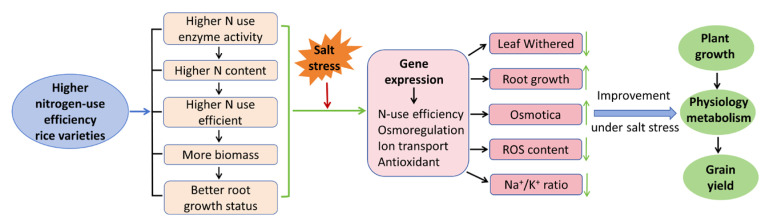
A presumed mechanism diagram of the way in which higher nitrogen-use efficiency increases salt stress tolerance in rice.

**Table 1 plants-14-00556-t001:** Genes and primer sequences used for qRT-PCR in this study.

Name	Primers Sequences (F: Forward R: Reverse)
*OsACT1*	F: TTCCAGCCTTCCTTCATA
	R: AACGATGTTGCCATATAGAT
*TOND1*	F: CCATGAGCTTCTCCTGCAGCT
	R: AGGAGCGCAGCTTGGAATCGT
*OsNPF6.1*	F: GGAGCGGCAAGATCGAGCACACG
	R: GAGGATGGCGAGGCAGGGGAAGAC
*OsP5CS1*	F: TGTGTACCAACGCGCTATGT
	R: TATATGCATCCACGGCGATA
*OsPDH1*	F: GCTACTGGGACTTGGGAGTG
	R: TCGATTGATACACCAATGTCTG
*OsCu/Zn-SOD*	F: TGTGACGGGACTTACTCCTGG
	R: CACCCATTCGTAGTATCGCCA
*OsCATB*	F: GCTGGTGAGAGATACCGGTCA
	R: TCAACCCACCGCTGGAGA
*OsAKT1*	F: AGAGATCCTTGATTCACTGCC
	R: TCTACTAACTCCACACTACCAG
*OsHKT1*	F: ACACCCAATATTATTCCTCTTAA
	R: CGGGAATACGCTAAAGG

## Data Availability

The data of this study are available from the corresponding author upon request.

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
