# Peer review of "Improvement in Nitrogen-Use Efficiency Increases Salt Stress Tolerance in Rice Seedlings and Grain Yield in Salinized Soil"

_plants, 2025, doi:10.3390/plants14040556_

Round 1

Reviewer 1 Report

Comments and Suggestions for Authors

Dear Authors, Thank you for submitting a significant manuscript on a very interesting topic, which you have described in great detail. I have a few minor observations. The abbreviations "SS" and "NIR" are used unnecessarily in the abstract. I recommend moving sentences 354-356 to the end of the penultimate paragraph (Line 87) of the introduction.

Author Response

Responses to Reviewer #1

Dear Authors, Thank you for submitting a significant manuscript on a very interesting topic, which you have described in great detail. I have a few minor observations. The abbreviations "SS" and "NIR" are used unnecessarily in the abstract. I recommend moving sentences 354-356 to the end of the penultimate paragraph (Line 87) of the introduction.

We appreciate the reviewer for the positive re-evaluation and for providing detailed comments and constructive suggestions. We have revised the related description and added a abbreviation in the manuscript for expressing the terms more clearly, and we revised our manuscript to address all the comments. We apologize for the English language problems and the manuscript had been edited by a professional English language editing service, meanwhile, we had proofread it to correct any possible mistakes. 

Reviewer 2 Report

Comments and Suggestions for Authors

The manuscript entitled " Improvement of nitrogen use efficient increased salt stress tolerance in rice seedlings and grain yield in salinized soil” is within the aim and scope of Plants.

I think the manuscript should be kind to make it easy for readers to understand. However, this manuscript needs a general correction, such as typos and incomprehensible sentences in English, explanations of abbreviations, and lack of information. So it's very difficult for me to understanding the manuscript. Therefore, it is required to resubmit the entire amended manuscript.

1. There are a lot of typos in English overall for publication. Need English editing

The tense of each verb used in the abstract, narrative, results, etc. is not appropriate

Also, many sentences are very difficult to understand in all manuscripts.

One example: What's the meaning of the sentence? “As shown in Fig. 1, salt stress induced the expression of TOND1 and OsNPF6.1, while the expression levels were decreased with the increase of salinity (line 104~105)

Furthermore, there are many overlapping and misspelled words.

One example: What is the purpose of your suggestion? “As shown the word ‘varietiesuthors’ in Discussion.

2. This manuscript has no explanation for the research results. So, information of genes (TOND1, OsNPF6.1, etc.) and rice varieties (D4, C9, J-667 etc.) is needed in the introduction.

3. What's the meaning of “CK”? distilled water (CK) or unpretreated control (CK). Describe the full name of the abbreviation and use a unified abbreviation. Many researchers use mock or control and provide additional explanations. For example: Control (C) is untreated with salt

4. Overall, figure legends and results should explain all the abbreviations used in the results.

Results (2.1) and All Figures: what is nitrogen efficient rice varieties? D4, C9, J-667 etc. This manuscript has no explanation. Also

Results (2.7) and Figures (7 and 8): what is SP?

5. Some indicated(specified) percentages which seem to be inaccurate need correction.

Examples: As shown in Fig. 5, In leaves, Na+/K+ ratio that NIR is higher than NER in SS2 and SS3. However, NIR SS1 was ‘22.0%’ which is lower than NER. Moreover, what is the correct percentage from ‘53.5.7%’ as shown in Fig. 6?

Comments on the Quality of English Language

There are a lot of typos in English overall for publication. Need English editing

The tense of each verb used in the abstract, narrative, results, etc. is not appropriate

Also, many sentences are very difficult to understand in all manuscripts.

Author Response

Responses to Reviewer #2

The manuscript entitled " Improvement of nitrogen use efficient increased salt stress tolerance in rice seedlings and grain yield in salinized soil” is within the aim and scope of Plants. I think the manuscript should be kind to make it easy for readers to understand. However, this manuscript needs a general correction, such as typos and incomprehensible sentences in English, explanations of abbreviations, and lack of information. So it's very difficult for me to understanding the manuscript. Therefore, it is required to resubmit the entire amended manuscript.

We appreciate the reviewer for the positive evaluation and for the critical and constructive comments and suggestions. We have revised our manuscript to address all the comments and suggestions. Below are our point-by-point responses.

Comment 1: There are a lot of typos in English overall for publication. Need English editing. The tense of each verb used in the abstract, narrative, results, etc. is not appropriate. Also, many sentences are very difficult to understand in all manuscripts. One example: What's the meaning of the sentence? “As shown in Fig. 1, salt stress induced the expression of TOND1 and OsNPF6.1, while the expression levels were decreased with the increase of salinity (line 104~105)”. Furthermore, there are many overlapping and misspelled words. One example: What is the purpose of your suggestion? “As shown the word ‘varietiesuthors’ in Discussion.

Our response: Thank you very much for your comment. We are sorry for the English language problems including those grammatical mistakes. We have had the manuscript edited by a professional English language editing service and have proofread it to correct any possible mistakes. The certificate is attached bellow.

Comment 2: This manuscript has no explanation for the research results. So, information of genes (TOND1, OsNPF6.1, etc.) and rice varieties (D4, C9, J-667 etc.) is needed in the introduction.

Our response: Thank you very much for your comment. We have added some description about the related genes and rice varieties in the introduction and results section, and more explanation for the results in the discussion section.

Comment 3: What's the meaning of CK? distilled water (CK) or unpretreated control (CK). Describe the full name of the abbreviation and use a unified abbreviation. Many researchers use mock or control and provide additional explanations. For example: Control (C) is untreated with salt.

Our response: Thank you very much for the comment. We apologize for the unclear description and have added a abbreviation in the manuscript for expressing the terms more clearly.

Comment 4: Overall, figure legends and results should explain all the abbreviations used in the results. Results (2.1) and All Figures: what is nitrogen efficient rice varieties? D4, C9, J-667 etc. This manuscript has no explanation. Also. Results (2.7) and Figures (7 and 8): what is SP?

Our response: Thank you very much for the comment. We apologize for the unclear description and have revised it.

Comment 5: Some indicated(specified) percentages which seem to be inaccurate need correction. Examples: As shown in Fig. 5, In leaves, Na+/K+ ratio that NIR is higher than NER in SS2 and SS3. However, NIR SS1 was 22.0% which is lower than NER. Moreover, what is the correct percentage from 53.5.7% as shown in Fig. 6?

Our response: We are very for the mistakes in the manuscript and we have carefully check the related value in the whole manuscript. With great thanks.

Comments on the Quality of English Language: There are a lot of typos in English overall for publication. Need English editing. The tense of each verb used in the abstract, narrative, results, etc. is not appropriate. Also, many sentences are very difficult to understand in all manuscripts.

Our response: We are sorry for the English language problems including those typos. The manuscript had received a professional English language editing service. In this revised version, we thoroughly checked and proofread every possible mistakes.

Again, thank you very much for your insightful comments and advices for our study. These comments are all valuable and helpful for improving our article. The whole revisions are shown in the document of Revised manuscript (changes are highlighted in red).

Reviewer 3 Report

Comments and Suggestions for Authors

Salinity stress represents a primary abiotic constraint on global crop productivity. This manuscript presents a systematic analysis of growth parameters, physiological metabolism, gene expression patterns, and yield components in rice cultivars with varying nitrogen use efficiency under salt stress conditions. The findings demonstrate that rice varieties with high nitrogen efficiency exhibit enhanced nitrogen utilization, improved salt stress tolerance, and increased crop yield. The manuscript demonstrates clear research objectives, comprehensive methodology, accurate result analysis, and thorough discussion. Here are some comments:

1. The annotation of Figure 1 needs to be clearer and more specific. In my understanding, the significance analysis in the figure refers to different treatments for the same variety, which needs to be clearly explained. The statistical significance indicators appear to contain errors and need verification. Similar concerns apply to Figures 4, 6, and 7.

2. Y-axis labels in Figure 4 require revision: 4A should read "Proline content in leaves" and 4D should read "Proline content in roots."

3. Although there is already a lot of data in the manuscript, what I want to know is why antioxidant enzyme activity was not measured.

4. Several analytical methodologies require more detailed descriptions.

5. Please pay attention to the correct writing of oxygen free radicals throughout the manuscript.

6. The primer sequences for β-actin should be included.

7. The conclusion is relatively simple, and it is recommended to expand it. It would be better if the author could draw a presumed mechanism diagram.

8. Line 90-93, this section is basically consistent with the conclusion. Importantly, these contents are not suitable for appearing in the introduction section.

Author Response

Responses to Reviewer #3

Salinity stress represents a primary abiotic constraint on global crop productivity. This manuscript presents a systematic analysis of growth parameters, physiological metabolism, gene expression patterns, and yield components in rice cultivars with varying nitrogen use efficiency under salt stress conditions. The findings demonstrate that rice varieties with high nitrogen efficiency exhibit enhanced nitrogen utilization, improved salt stress tolerance, and increased crop yield. The manuscript demonstrates clear research objectives, comprehensive methodology, accurate result analysis, and thorough discussion. Here are some comments:

Sincerely, thanks for spending valuable time in re-assessing our manuscript, and we appreciate your comments and suggestions very much, which are very valuable and helpful to improve this manuscript. Following your advices, we have carefully revised the MS point-to-point. In addition, the manuscript had received a professional English language editing service. In this revised version, we thoroughly checked and proofread every possible mistakes. Below are our point-by-point responses.

Comment 1: The annotation of Figure 1 needs to be clearer and more specific. In my understanding, the significance analysis in the figure refers to different treatments for the same variety, which needs to be clearly explained. The statistical significance indicators appear to contain errors and need verification. Similar concerns apply to Figures 4, 6, and 7.

Our response: Thank you very much for your comment. We apologize for the mistake, and have checked the whole figures and revised it.

Comment 2: Y-axis labels in Figure 4 require revision: 4A should read "Proline content in leaves" and 4D should read "Proline content in roots.".

Our response: We have revised the figures and the related legends, with great thanks.

Comment 3: Although there is already a lot of data in the manuscript, what I want to know is why antioxidant enzyme activity was not measured.

Our response: Thank you for your comment. In this manuscript, we measured the osmotic adjustment indexes, Na+/K+ content and ROS content in different rice varieties, which indicated the osmotic stress, ion toxicity and oxidative stress induced by salt stress. Therefore, we evaluated the salt stress tolerance of different nitrogen use efficiency rice varieties based on the above three factors, rather than one of them. In addition, we measured the ROS content to analyze the oxidative stress degree under salt stress, and the expression levels of antioxidant enzyme related genes to analyze the antioxidant defenses capacity in different rice varieties. With great thanks.

Comment 4: Several analytical methodologies require more detailed descriptions.

Our response: We have added more detailed descriptions about several indexes, with great thanks.

Comment 5: Please pay attention to the correct writing of oxygen free radicals throughout the manuscript.

Our response: Thanks for your comment. We have revised it.

Comment 6: The primer sequences for β-actin should be included.

Our response: Thanks for your comment. In this manuscript, the housekeeping gene β-actin (GenBank ID: X15865.1) was used as an internal standard, according to the previous studies (Yu et al. 2015; Zhang et al. 2017; Liu et al. 2022), it was also named OsACT1. The primer sequences for β-actin was showed in Table 1.

  • Yu H, Jiang W, Liu Q, et al. Expression pattern and subcellular localization of the ovate protein family in rice. PLoS ONE, 2015, 10:e0118966.
  • Zhang H, Liu X, Zhang R, et al. Root damage under alkaline stress is associated with reactive oxygen species accumulation in rice (Oryza sativa). Front. plant Sci. 2017, 8: 1580.
  • Liu XL, Xie XZ, et al. RNAi-mediated suppression of the abscisic acid catabolism gene OsABA8ox1 increases abscisic acid content and tolerance to saline-alkaline stress in rice (Oryza sativa). Crop J. 2022, 10: 354-367.

Comment 7: The conclusion is relatively simple, and it is recommended to expand it. It would be better if the author could draw a presumed mechanism diagram.

Our response: Thank you for your useful comments and we have added a presumed mechanism diagram to explain the research results.

Comment 8: Line 90-93, this section is basically consistent with the conclusion. Importantly, these contents are not suitable for appearing in the introduction section.

Our response: Thanks for your comment. We have revised the related description and removed it in the conclusion section.

Again, thank you very much for spending valuable time reading our manuscript. These comments are all valuable and helpful for improving our article. The whole revisions are shown in the document of Revised manuscript (changes are highlighted in red).

Round 2

Reviewer 2 Report

Comments and Suggestions for Authors

The author accepted my review and faithfully reflected its contents in the revised version. so, I think this manuscript can be published in this journal.